# Optimizing Active Tuberculosis Case Finding: Evaluating the Impact of Community Referral for Chest X-ray Screening and Xpert Testing on Case Notifications in Two Cities in Viet Nam

**DOI:** 10.3390/tropicalmed5040181

**Published:** 2020-11-30

**Authors:** Tuan Huy Mac, Thuc Huy Phan, Van Van Nguyen, Thuy Thu Thi Dong, Hoi Van Le, Quan Duc Nguyen, Tho Duc Nguyen, Andrew James Codlin, Thuy Doan To Mai, Rachel Jeanette Forse, Lan Phuong Nguyen, Tuan Ho Thanh Luu, Hoa Binh Nguyen, Nhung Viet Nguyen, Xanh Thu Pham, Phap Ngoc Tran, Amera Khan, Luan Nguyen Quang Vo, Jacob Creswell

**Affiliations:** 1Hai Phong Lung Hospital, Hai Phong 180000, Vietnam; drtuanhpl@gmail.com (T.H.M.); tholaohp@gmail.com (T.D.N.); 2Provincial Department of Health, Hai Phong 180000, Vietnam; thucphanhuy@gmail.com (T.H.P.); nguyenyquanhp@gmail.com (Q.D.N.); phamthuxanh@gmail.com (X.T.P.); 3Provincial Department of Health, Quang Nam 560000, Vietnam; nhivan6@gmail.com; 4Friends for International TB Relief, Ha Noi 100000, Vietnam; thuy.dong@tbhelp.org (T.T.T.D.); andrew.codlin@tbhelp.org (A.J.C.); thuy.mai@tbhelp.org (T.D.T.M.); rachel.forse@tbhelp.org (R.J.F.); lan.nguyen@tbhelp.org (L.P.N.); 5Viet Nam National Lung Hospital, Ha Noi 100000, Vietnam; hoilv@yahoo.com (H.V.L.); nguyenbinhhoatb@yahoo.com (H.B.N.); vietnhung@yahoo.com (N.V.N.); 6Clinton Health Access Initiative, Ha Noi 100000, Vietnam; tluu@clintonhealthaccess.org; 7Pham Ngoc Thach Quang Nam Hospital, Quang Nam 560000, Vietnam; phapqnam@gmail.com; 8Stop TB Partnership, 1218 Geneva, Switzerland; amerak@stoptb.org (A.K.); jacobc@stoptb.org (J.C.); 9Interactive Research and Development, Singapore 189677, Singapore

**Keywords:** tuberculosis, active case finding, community health workers, mobile X-ray screening

## Abstract

To accelerate the reduction in tuberculosis (TB) incidence, it is necessary to optimize the use of innovative tools and approaches available within a local context. This study evaluated the use of an existing network of community health workers (CHW) for active case finding, in combination with mobile chest X-ray (CXR) screening events and the expansion of Xpert MTB/RIF testing eligibility, in order to reach people with TB who had been missed by the current system. A controlled intervention study was conducted from January 2018 to March 2019 in five intervention and four control districts of two low to medium TB burden cities in Viet Nam. CHWs screened and referred eligible persons for CXR to TB care facilities or mobile screening events in the community. The initial diagnostic test was Xpert MTB/RIF for persons with parenchymal abnormalities suggestive of TB on CXR or otherwise on smear microscopy. We analyzed the TB care cascade by calculating the yield and number needed to screen (NNS), estimated the impact on TB notifications and conducted a pre-/postintervention comparison of TB notification rates using controlled, interrupted time series (ITS) analyses. We screened 30,336 individuals in both cities to detect and treat 243 individuals with TB, 88.9% of whom completed treatment successfully. All forms of TB notifications rose by +18.3% (95% CI: +15.8%, +20.8%). The ITS detected a significant postintervention step-increase in the intervention area for all-form TB notification rates (IRR(β6) = 1.221 (95% CI: 1.011, 1.475); *p* = 0.038). The combined use of CHWs for active case findings and mobile CXR screening expanded the access to and uptake of Xpert MTB/RIF testing and resulted in a significant increase in TB notifications. This model could serve as a blueprint for expansion throughout Vietnam. Moreover, the results demonstrate the need to optimize the use of the best available tools and approaches in order to end TB.

## 1. Introduction

In 2014, the Government of Viet Nam passed legislation with the vision of ending tuberculosis (TB) by 2030 [1]. This goal seemed within reach, with a reported treatment coverage of 87% in 2018 [2]. However, the second national prevalence survey revealed that a large detection gap remained as treatment coverage estimates were revised downwards to 57% in 2019 [3]. This gap, in human terms, means that approximately 74,000 people with TB are unreached by the National TB Control Program (NTP), and this is a formidable barrier to the nation’s ambitions to end TB.

The prevalence survey results also demonstrated that a “business-as-usual” approach was insufficient to end TB and evinced the need for active case finding (ACF), as many people with TB identified in the survey were previously undiagnosed. To generate the evidence in support of an ACF scale-up, Viet Nam conducted a randomized controlled trial that featured population-wide outreach with dedicated teams conducting door-to-door visits to offer unrestricted testing on the Xpert MTB/RIF (Xpert) assay as the initial diagnostic test on any obtainable expectorate [4]. While the trial demonstrated the possibility of drastic reductions in TB prevalence through ACF, the approach was extremely resource-intensive.

The programmatic implementation of resource-intensive strategies, even if strongly evidenced, has traditionally faced challenges [5,6,7]. Common barriers are unsustainable human, financial and technical resource requirements [8]. In Viet Nam, these barriers are exacerbated by the resource needs to compensate for lower-than-expected treatment coverage following the second prevalence survey as well as delays in the transition of TB program financing to national social health insurance [9]. Hence, the NTP is keen to optimize its resource utilization by providing Xpert testing mainly to persons with the highest risk of suffering from active TB disease.

There is extensive evidence that parenchymal abnormalities suggestive of TB on chest radiography (CXR) represent a critical risk factor of active TB disease [10,11]. Based on this evidence, CXR has been recommended as a screening and triage tool placed early in TB diagnostic algorithms [12]. The NTP has committed to expand the use of CXR as a triage tool for the indication of Xpert testing as part of its national diagnostic roadmap [9]. This diagnostic algorithm, locally dubbed the “Double-X” or “2X” algorithm, is cited to have the potential to significantly reduce laboratory burdens, costs per true case detected and false-positive bacteriological test results [13,14]. Previous studies have also found that this algorithm offers a good balance between sensitivity and costs [15,16].

There is a growing consensus that a comprehensive, intensified approach is needed to progress towards ending TB [17]. Critical components of such a comprehensive approach include ACF and prompt treatment initiation as a means to stop transmission, treating subclinical TB infection for persons most likely to progress to active TB disease and addressing the social determinants of TB treatment [18,19,20]. In 2017, the NTP launched the Zero TB Viet Nam (ZTV) project as a pilot for such an approach in Viet Nam. As part of its ACF activities, the project tested an intensified TB case finding approach with a targeted case detection through the engagement of community health workers (CHW) as well as an early diagnosis and initiation of appropriate treatment via the 2X diagnostic algorithm [21].

## 2. Materials and Methods

### 2.1. Study Design

This study was a prospective controlled cohort intervention study that aimed to measure the yield, treatment outcomes and additional impact on case notifications of an intensified community-based active case finding intervention.

### 2.2. Study Setting

The study was conducted from January 2018 until March 2019 in nine districts in Viet Nam. These included five intervention and four control districts (Figure 1).

The intervention area included four districts in Hai Phong provincial city and the district-level city of Hoi An in the Quang Nam province. The five intervention districts had a population of 717,343 and notified 376 people with all forms TB in 2017 for a case notification rate (CNR) of 52/100,000. The control area was selected based on recommendations from the provincial TB programs to match the intervention areas in approximate size and TB burden as well as the absence of any case finding interventions. The control area included three districts in Hai Phong and the district-level city of Tam Ky in Quang Nam. The cumulative control district population was 377,130 and notified 150 people with all forms TB in 2017 for a CNR of 40/100,000. The District TB Unit (DTU) and other NTP-affiliated TB clinics managed diagnosis, treatment and notification according to national guidelines under the technical supervision of the Hai Phong and Quang Nam provincial lung hospitals.

### 2.3. Community Health Workers

The study recruited a cadre of 60 CHWs in both cities to carry out ACF activities. The community network engaged in Hai Phong and Hoi An was provided by the General Department of Population and Family Planning (GDPFP) under the Ministry of Health. The GDPFP employs a network of over 11,000 commune coordinators, who are full-time government staff responsible for coordinating a government network of community volunteers at the subcommune level. Their primary duties consist of population surveillance as well as advocacy and care in the fields of sexual and reproductive health, and maternal and child health [22]. Most coordinators hold intermediate college degrees in the fields of midwifery and nursing. The 60 individuals selected by the study to be CHWs received a two-day training focused on a basic level of TB literacy, core study objectives and responsibilities, and associated recording and reporting requirements on the study’s Android-based, mobile health (mHealth) application. Each CHW received a remuneration package that included a stipend of USD 22.70 per month as well as performance-based incentives of USD 0.45 per successful CXR referral and USD 2.27 per successful linkage to treatment of a person diagnosed with TB.

### 2.4. Target Populations

The target groups consisted of contacts of a TB index case and urban priority groups, including people aged 55 years and over, individuals with a history of TB treatment during the prior two years (2016–2017) and urban poor households based on national poverty definitions. These groups were prioritized at the discretion of the Provincial TB control programs based on the historically observed higher incidence in these groups.

Index cases were defined as anyone prospectively notified with drug-susceptible TB regardless of bacteriological status or disease site (all forms). Anyone who shared a kitchen with the index case or close contacts who had interacted with the index case at least once per week met the inclusion criteria for contact investigation. CHWs obtained index patient details from NTP patient registers. Household contacts (HHC) were enumerated and screened during home visits. Index patients were requested to identify other close contacts and provide relevant phone numbers and/or addresses. Elderly individuals were identified through the GDPFP’s local census. Low income households were identified by local authorities.

### 2.5. ACF Activities

The study’s process flow of the ACF activities is depicted in Figure 2. CHWs systematically screened target populations though home visits and mobile screening events using a bespoke mHealth application (TechUp/Clinton Health Access Initiative; Ha Noi, Viet Nam). Screening activities were promoted and enhanced through the distribution of educational pamphlets to encourage awareness and participation. During home visits, CHWs verbally screened all individuals for clinical symptoms and a history of TB. Clinical symptoms included (productive) cough, hemoptysis, fever, weight loss, night sweats, dyspnea, chest pain and fatigue, as per national guidelines. Any household contact was eligible for CXR referral, irrespective of symptomatic presentation. All other individuals targeted for screening were eligible for CXR if they reported at least one symptom or had a history of TB. Eligible individuals received a voucher for a free CXR at NTP-designated facilities. In addition to the home visits, the provincial TB programs organized 46 community CXR screening events in Hai Phong and four in Hoi An using mobile CXR vans to improve uptake. CXR images were read by a NTP-trained radiologist, and persons with abnormalities suggestive of TB provided a sputum sample for testing on the Xpert MTB/RIF assay (Cepheid; Sunnyvale, CA, USA). Any symptomatic individual without CXR results or with chest radiograph findings not associated with TB was referred for smear microscopy. Persons with a positive Xpert result and a history of TB were further tested on smear and culture, if needed, as per NTP guidelines. Symptomatic persons with negative sputum test results were evaluated for a clinical diagnosis according to NTP guidelines. Children with presumptive TB were referred to the Provincial Lung Hospital for further evaluation. Persons diagnosed with active TB were linked to care at their provider of choice and followed up until treatment completion.

### 2.6. Data Analysis

We calculated the number and proportion of persons in each step in the TB care cascade starting with those verbally screened and eligible for CXR until treatment completion [23] by city and further bifurcated the cascade by CXR screening site, i.e., facility vs. mobile van. We calculated the cumulative and city-disaggregated changes in all forms and bacteriologically confirmed TB notifications applying historical and contemporaneous controls [24]. To verify these results in light of secular trends, we conducted comparative interrupted time series (ITS) analyses of population-standardized aggregated quarterly all-form and bacteriologically confirmed TB notification rates. The ITS analyses employed segmented methods to model a postintervention step-change (β_6_) and subsequent trend differences (β_6_). The time series data consisted of 21 observations of quarterly TB notification rates in intervention and control areas. This included 16 preintervention quarters and five quarters of implementation of ACF interventions [25]. These methods were applied to marginal log-linear Poisson regression models using the generalized estimating equation (GEE) approach. We tested for serial autocorrelation using the Cumby–Huizinga test with a threshold of *p* < 0.05, and specified lag parameters of the model based on the lowest quasi-likelihood information criterion values. Statistical analyses were performed on Stata 13 (StataCorp; College Station, TX, USA). Hypothesis tests were two-sided, and point estimates included 95% confidence intervals.

### 2.7. Ethical Considerations

The study protocol was approved by the Scientific and Ethics Committee of the Hai Phong University of Medicine and Pharmacy (82/QD-YDHP) and the Ethics Committee for Biomedical Research & Committee for Science and Technology of Pham Ngoc Thach Hospital (430/HDDD-PNT). The NTP (909/QD-BVPTW) and the Ministry of Health (3651/QĐ-BYT) granted technical and administrative approval for the study’s implementation. We obtained written informed consent from participants during each screening encounter and de-identified all personal data. Persons who did not consent still received testing and treatment per the study protocol and NTP guidelines, but were excluded from all analyses.

## 3. Results

### 3.1. ACF Outputs

The aggregate TB care cascade is shown in Figure 3. Over 15 months, the CHWs verbally screened 30,336 individuals. CXR results were recorded for 67.2% (20,389/30,336), among whom the abnormality rate was 18.4% (3749/20,389). We tested the sputum of 2249 individuals, including 1655 with Xpert tests (44.1% of those eligible) and 594 smear tests. We diagnosed 268 people with TB. Of these individuals with TB, 90.7% (243/268) initiated treatment. This represents a yield of 801/100,000 and an NNS of 125. Of these people with TB, 88.9% (216/243) completed treatment successfully, 7.8% (19/243) were lost to follow-up, 1.6% (4/243) were not evaluated due to a transfer to an MDR-TB treatment regimen and 1.6% (4/243) died.

Table 1 shows the disaggregated TB care cascade and treatment outcomes by the three urban priority groups. Of the 30,336 individuals screened, 14.0% (4259/30,336) were HHCs, among whom 41 were initiated on TB treatment for a prevalence of 963/100,000 and an NNS of 104. Close contacts comprised just 4.3% (1313/30,336) of the sample. Eight close contacts were initiated on treatment for a prevalence of 609/100,000 or an NNS of 164. The proportion of urban priority area residents among screened individuals was 82.1% (24,764/30,336), of whom 194 individuals were initiated on treatment for a prevalence of 783/100,000 and an NNS of 128. Overall, 88.9% (216/243) of those enrolled successfully completed treatment (Table 2). The treatment success rates among household contacts, social and close contacts, and urban priority area residents were 87.8% (36/41), 100.0% (8/8) and 88.7% (172/194), respectively.

Figure 4 shows the TB care cascade disaggregated by city and by facility-based or community-based CXR screening location. There were notable differences between the two cities. In Hai Phong, CHWs engaged and verbally screened 23,967 persons from the target populations, of whom 71.7% (17,191/23,967) presented for a CXR. In comparison, only 50.2% (3198/6369) of the individuals screened in Hoi An had a recorded CXR result. Of the total number of CXRs taken in Hai Phong, only 19.9% (3428/17,191) were facility-based, while 80.1% (13,763/17,191) of CXRs originated from mobile CXR screening events. In comparison, 71.2% (2278/3198) of CXRs in Hoi An were taken at a facility, and only 28.8% (920/3198) of CXRs were recorded from mobile screening events. Similarly, the aggregate proportion of Xpert testing among all sputum tests in Hai Phong was 85.0% (1516/1783) compared to 29.8% (139/466) in Hoi An. ACF activities in Hai Phong yielded 223 persons diagnosed with TB, of whom 88.8% (198/223) were initiated on treatment, corresponding to a yield of 826/100,000 and an NNS of 121. In Hoi An, ACF activities resulted in 45 persons diagnosed with TB and initiated on treatment, corresponding to a yield of 707/100,000 and an NNS of 142.

### 3.2. Impact on Notifications

All forms TB notifications across both intervention cities (Table 3) increased by +18.3% (+15.8%, +20.8%), corresponding to 165 (142, 188) additional TB notifications. All-form TB notifications by city rose by +11.0% (+9.2%, +12.9%) in Hai Phong compared to +35.4% (+26.8%, +44.1%) in Hoi An. These rates corresponded to 123 (102, 144) additional all-form TB notifications in Hai Phong and 42 (32, 52) in Hoi An. Bacteriologically-confirmed TB notifications increased by +32.9% (+27.8%, +37.9%), corresponding to 108 (91, 125) additional TB cases over the baseline. In Hai Phong, the estimated impact on notifications was +30.6% (+24.9%, +36.3%), corresponding to 76 (62, 90) additional cases. In Hoi An, the increase in notifications was +36.5% (+26.4%, +46.5%), corresponding to 32 (23, 41) additional people with bacteriologically confirmed TB.

The ITS analyses results are in Table 4 and Figure 5. The baseline median quarterly all-form TB notification rate was 24.7 (IQR: 21.9–27.1) per 100,000 in the intervention area and 22.0 (IQR: 19.1–23.6) in the control area. In the post-implementation period, there was a significant step-change in all-form TB notification rates (IRR(β_6_) = 1.221 (1.011, 1.475); *p* = 0.038) and bacteriologically confirmed TB notification rates (IRR(β_6_) = 1.535 (1.067, 2.210); *p* = 0.021).

## 4. Discussion

The results of our study demonstrate that CHWs were effective in mobilizing and verbally screening a large number of people and in ultimately detecting persons suffering from TB. The analysis of notification data showed that a substantial portion of this yield translated to additional TB notifications over the baseline. Adjusting for demographic changes and other secular confounders showed that the additional TB notifications comprised a significant change from the status quo. Our results add to a growing literature documenting the impact of community-based ACF on increasing TB case notifications [26,27].

The yield achieved on this project underlines the high number of people in the community who have TB yet remain unreached and were underestimated prior to Viet Nam’s second prevalence survey. In this study, this was particularly evident among urban priority area residents without documented exposure to an index patient. A key catalyst for the high yield was the use of the 2X diagnostic algorithm and the use of Xpert as the initial diagnostic tool. Prior to this project, Hai Phong and Hoi An used smear as the primary diagnostic tool, while Xpert was reserved for drug-susceptibility testing. The impact of using Xpert for TB diagnosis on the yield was evidenced by the intercity differences in the number of people tested and bacteriologically confirmed on Xpert. Specifically, the Hai Phong provincial TB program was able to avail Xpert to a larger proportion of people screened and as such achieved a higher yield. This elevated yield from the introduction of Xpert as the initial diagnostic test has been previously documented in many other settings [28,29,30].

However, studies have shown that a high yield and low NNS from more sensitive diagnostic tools and algorithms in isolation does not necessarily translate to additional notifications [31,32]. As vulnerable groups with low NNS are often also small in numbers, it has been noted that the use of more sensitive diagnostic tools and algorithms must be complemented by efforts to increase the number of people referred for screening and testing [33]. This particularly applies to individuals who may otherwise not have sought care [34] due to the lack of symptomatic presentation or recognition, challenging socioeconomic conditions or indomitable sociocultural barriers [35,36,37,38]. One way of reaching more people with TB is effective community engagement.

In this study, community engagement was facilitated by the GDPFP network of CHWs. These CHWs were able to seamlessly integrate the study’s ACF activities into their daily responsibilities. As Viet Nam’s population growth and demographics have stabilized, the GDPFP is looking to repurpose its community network to address other public health concerns. This type of repurposing of existing health worker networks has also been successfully implemented in other settings [39]. As such, the study raised notifications without creating new or redundant community structures and thereby offers a potential avenue for a national scale-up. Accessing the existing GDPFP network played an important role in the fidelity of this study [40]. Coordination with the provincial branches of the GDPFP generated operational efficiencies by simplifying the recruitment process, given that GDPFP coordinator rosters were readily available. Since these coordinators were experienced in the delivery of public health services, it was possible to impart TB literacy in a relatively short amount of time at a high level of quality [40]. As these coordinators were full-time employees, attrition was also lower than what was typically observed in community health worker and volunteer networks [41,42].

The synergies of combining responsibilities likely also contributed to the high efficiency in our study. Both cities exhibited a similar NNS that was relatively low in comparison to reported values from systematic screening in other target groups [43]. Similarly, our study recorded a high rate of CXR screening among persons that were verbally engaged. According to field staff, the CHWs leveraged their existing relationships with community leaders to establish a personal referral system. Through this system, symptomatic persons could confide in a trusted community member before presenting at a health facility. This trust in CHWs is a key reason why their services are often considered more person-centric than that of formal healthcare providers [44]. A critical question consists in the ability of repurposed CHWs to multitask without making concessions on the quality of care. While we did not systematically measure the level of multitasking and performance across all of the individual CHWs’ duties or the CHWs’ contribution to the measured treatment outcomes on this study, the ability of CHWs to effectively manage multiple responsibilities for TB and other public health indications has been documented elsewhere [45,46,47]. For example, Ethiopian health workers reported receiving positive feedback from their communities on the provision of multiple tasks and noticed the impact that their work made, which was motivating for them [48].

Nevertheless, we still faced challenges in community mobilization efforts despite the support from the CHWs. Specifically, we found that CXR triaging could work as an access barrier to care as health-seeking persons had to present at an X-ray facility. This risk is noted in the WHO guidelines on the role of chest radiography in TB [13]. Studies have also linked facility-based health-seeking to higher directly incurred expenses and lost time [49]. To reduce these diagnostic access barriers, the project implemented weekend community screening events using mobile CXR vans. These vans are abundant in the Vietnamese context for statutory occupational health screening in formal workplace settings and have been used in the past for TB screening in closed settings and on prevalence surveys [50]. The benefit of this strategy was evident in Hai Phong, where the participation rate in CXR screening was substantially higher than in Hoi An. The consequence of the higher CXR participation rate was that more health-seeking persons were able to access Xpert testing. The utility of mobile CXR events and their ability to reduce TB care access barriers, particularly for vulnerable populations such as the urban priority area residents, has also been demonstrated in other settings [51,52]. Another limitation of the approach consisted in the potential heterogeneity in CXR interpretations across different human readers. Future studies should further explore this limitation and the utility of computer-aided reading solutions in active case finding settings such as the one implemented in this project [53].

Our study was limited by confounding effects in the control districts of Hai Phong from a concurrent scale-up of public-private mix (PPM) activities in the city [54]. This resulted in a positive notification trend in the control districts, which affected the additionality results. This PPM scale-up was also evident in the proportion of TB patients detected by our intervention who chose private sector treatment. The ITS analysis detected a significant step-change but did not detect a change in long-term trends, suggesting diminishing returns from our ACF activities. This lack of sustained impact may have been a result of the short timeframe of the project and should be further investigated through appropriately powered cluster randomized trials. Moreover, the additional notifications constituted only half of the estimated case detection gap, suggesting that a more comprehensive evaluation, inclusive of ACF and PPM activities, is required to close the TB treatment coverage gap.

Last, the study focused mainly on active tuberculosis, and future studies should investigate the possibility of integrated screening, testing and treatment of TB and latent TB infection. Similarly, as high-burden countries begin to expand Xpert coverage through strategies such as the 2X diagnostic algorithm in Viet Nam, the cost-effectiveness of these strategies need further investigation to complement their potential epidemiologic benefits.

## 5. Conclusions

Our study evidenced that ACF, through a combination of community engagement through CHWs and the reduction of access barriers to CXR screening and subsequent Xpert testing, can locate previously unreached persons with TB. These findings may serve as further evidence in favor of the scale-up of intensified case finding activities for TB in Viet Nam and other high-burden settings.

## Figures and Tables

**Figure 1 tropicalmed-05-00181-f001:**
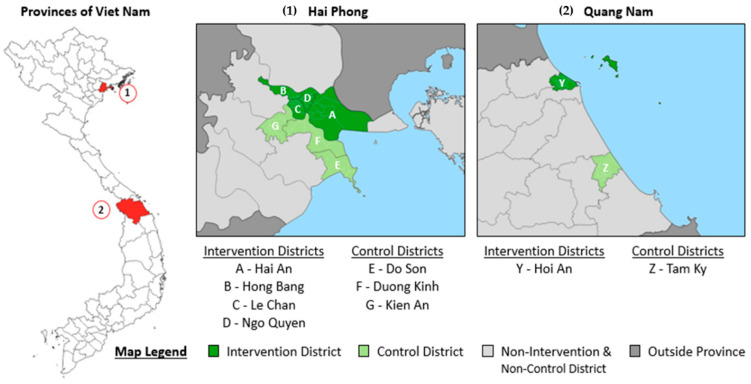
Map of intervention and control provinces and districts. Hai Phong and Quang Nam, Viet Nam. January 2018 to March 2019.

**Figure 2 tropicalmed-05-00181-f002:**
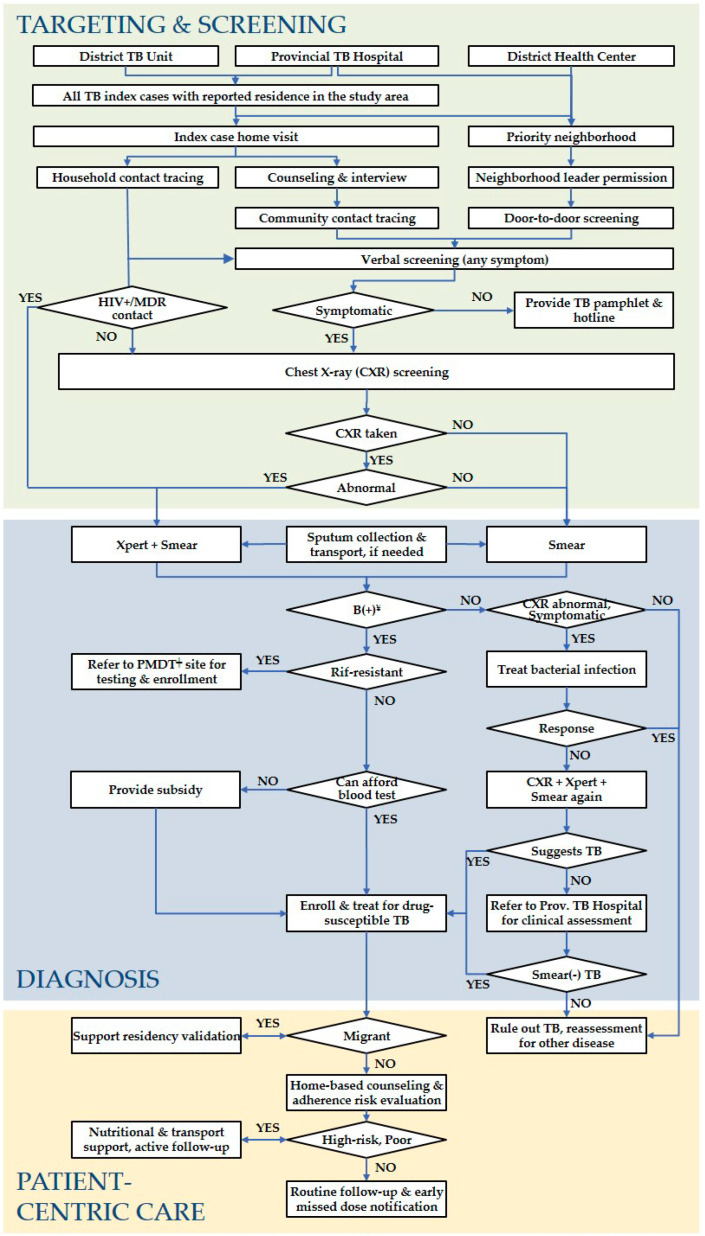
Process flow of the ACF activities. Hai Phong and Quang Nam, Viet Nam. January 2018 to March 2019. ¥ Bacteriologically-confirmed; ‡ Programmatic Management of Drug-Resistant TB.

**Figure 3 tropicalmed-05-00181-f003:**
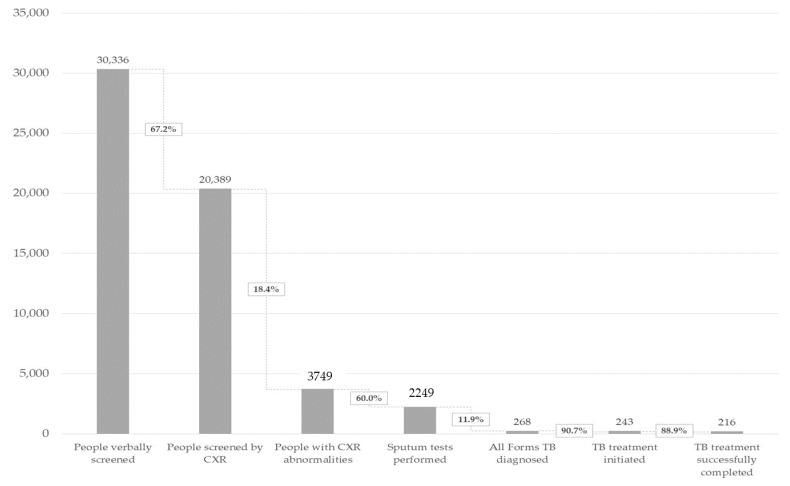
Aggregate TB care cascade. Hai Phong and Quang Nam, Viet Nam. January 2018 to March 2019.

**Figure 4 tropicalmed-05-00181-f004:**
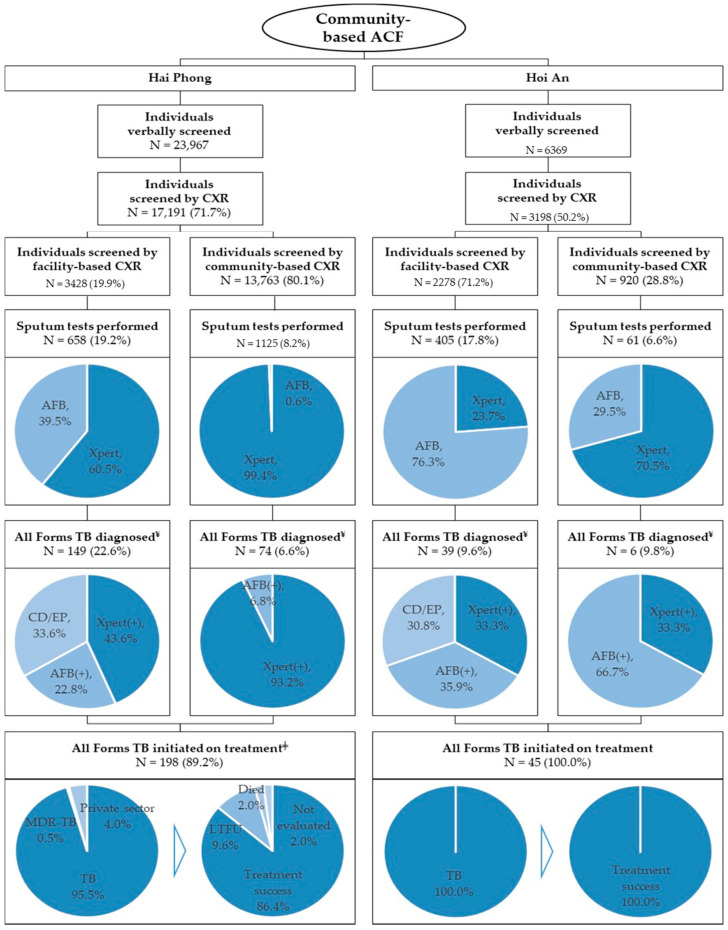
TB care cascade by city and CXR screening site. Hai Phong and Quang Nam, Viet Nam. January 2018 to March 2019. ¥ In the event that a positive AFB and Xpert result was recorded, the patient was categorized as an AFB(+) case. ‡ All private sector patients had rifampicin-susceptible TB.

**Figure 5 tropicalmed-05-00181-f005:**
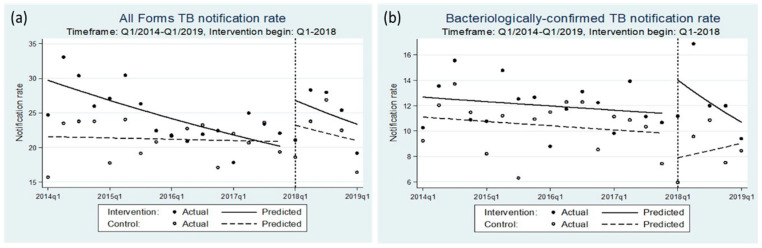
Comparative ITS analysis model graphs of population-standardized quarterly notification rates of (**a**) all-form TB case notification rates and (**b**) bacteriologically confirmed TB case notification rates for intervention vs. control areas. Hai Phong and Quang Nam, Viet Nam. January 2018 to March 2019. Notes: In the baseline period, the predicted line indicates the fitted model based on historical actual notification rates.

**Table 1 tropicalmed-05-00181-t001:** TB care cascade disaggregated by urban priority group. Hai Phong and Quang Nam, Viet Nam. January 2018 to March 2019.

	Total N (%)	Household Contacts N (%)	Social & Close Contacts N (%)	Urban Priority Area Residents N (%)
Individuals verbally screened	30,336 (100.0%)	4259 (100.0%)	1313 (100.0%)	24,764 (100.0%)
Individuals screened by CXR	20,389 (67.2%)	2087 (49.0%)	563 (42.9%)	17,739 (71.6%)
--Individuals with abnormal CXR screen	3749 (12.4%)	266 (6.2%)	101 (7.7%)	3382 (13.7%)
Individuals tested for TB (any sputum test)	2249 (7.4%)	184 (4.3%)	65 (5.0%)	2000 (8.1%)
--Individuals tested for TB with Xpert	1655 (5.5%)	120 (2.8%)	45 (3.4%)	1490 (6.0%)
Individuals diagnosed with All Forms TB	268 (0.9%)	44 (1.0%)	9 (0.7%)	215 (0.9%)
--Individuals diagnosed Xpert(+)	149 (0.5%)	14 (0.3%)	8 (0.6%)	127 (0.5%)
All Forms TB patients started on treatment	243 (0.8%)	41 (1.0%)	8 (0.6%)	194 (0.8%)
--NNS	125	104	164	128

**Table 2 tropicalmed-05-00181-t002:** TB treatment outcomes by urban priority group. Hai Phong and Quang Nam, Viet Nam. January 2018 to March 2019.

	Total N (%)	Household Contacts N (%)	Social & Close Contacts N (%)	Urban Priority Area Residents N (%)
Treated successfully	216 (88.9%)	36 (87.8%)	8 (100.0%)	172 (88.7%)
Lost to follow-up	19 (7.8%)	5 (12.2%)	0 (0.0%)	14 (7.2%)
Died	4 (1.6%)	0 (0.0%)	0 (0.0%)	4 (2.1%)
Not evaluated/failure	4 (1.6%)	0 (0.0%)	0 (0.0%)	4 (2.1%)

**Table 3 tropicalmed-05-00181-t003:** Impact analysis [24] of all forms and bacteriologically confirmed TB notifications by city. Hai Phong and Quang Nam, Viet Nam. January 2018 to March 2019.

	Cumulative Notifications		Trend Differences			
	Baseline Period ^†^	Intervention Period	^#^ Cases	95% CI	% Change ^§^	95% CI
**All forms TB**						
Cumulative additional notifications			165	(142,188)	18.3%	(15.8%, 20.8%)
Hai Phong			123	(102,144)	11.0%	(9.2%, 12.9%)
Intervention area	706	850	144	(123,165)	20.4%	(17.4%, 23.4%)
Control area	224	245	21	(12,30)	9.4%	(5.6%, 13.2%)
Hoi An			42	(32,52)	35.4%	(26.8%, 44.1%)
Intervention area	112	148	36	(26,46)	32.1%	(23.5%, 40.8%)
Control area	182	176	−6	(–11,–1)	−3.3%	(−5.9%, −0.7%)
**Bacteriologically confirmed TB**						
Cumulative additional notifications			108	(91,125)	32.9%	(27.8%, 37.9%)
Hai Phong			76	(62,90)	30.6%	(24.9%, 36.3%)
Intervention area	354	419	65	(51,79)	18.4%	(14.3%, 22.4%)
Control area	90	79	−11	(–17,–5)	−12.2%	(−19.0%, −5.5%)
Hoi An			32	(23,41)	36.5%	(26.4%, 46.5%)
Intervention area	77	93	16	(9,23)	20.8%	(11.7%, 29.8%)
Control area	102	86	−16	(–23,–9)	−15.7%	(−22.7%, −8.6%)

^†^ The baseline period consists of the January 2017–December 2017 timeframe; the cumulative baseline notifications are the sum of notifications matched by quarter to the intervention period of January 2018–March 2019 to account for seasonality, i.e., Q1 2018 matched with Q1 2017, Q2 2018 matched with Q2 2017, Q3 2018 matched with Q3 2017, Q4 2018 matched with Q4 2017 and Q1 2019 matched with Q1 2017. ^§^ The sums of the percentage point estimates include rounding effects; The number of cases denotes the double difference between pre- and postimplementation and between intervention and control areas.

**Table 4 tropicalmed-05-00181-t004:** Comparative ITS analysis model parameters of population-standardized quarterly notification rates of all-form and bacteriologically confirmed TB cases for intervention vs. control districts ^¥^. Hai Phong and Quang Nam, Viet Nam. January 2018 to March 2019.

Comparative ITS Analysis Model Parameters	Intervention vs. Control Districts		
	IRR ^‡^	95% CI	*p*-Value ^þ^
**All Forms TB**			
Baseline rate (*β*_0_)	21.563	(20.108, 23.124)	<0.001
Preintervention trend, control (*β*_1_)	0.998	(0.990, 1.006)	0.590
Postintervention step change, control (*β*_2_)	1.116	(0.952, 1.308)	0.178
Postintervention trend, control (*β*_3_)	0.977	(0.923, 1.034)	0.427
Difference in baseline (*β*_4_)	1.378	(1.270, 1.495)	<0.001
Difference in preintervention trends (*β*_5_)	0.977	(0.967, 0.986)	<0.001
Difference in postintervention step change (*β*_6_)	1.221	(1.011, 1.475)	0.038
Difference in postintervention trends (*β*_7_)	1.015	(0.948, 1.086)	0.676
**Bacteriologically confirmed TB**			
Baseline rate (*β*_0_)	11.107	(9.562, 12.901)	<0.001
Preintervention trend, control (*β*_1_)	0.992	(0.975, 1.009)	0.361
Postintervention step change, control (*β*_2_)	0.807	(0.587, 1.109)	0.186
Postintervention trend, control (*β*_3_)	1.043	(0.935, 1.163)	0.448
Difference in baseline (*β*_4_)	1.141	(0.956, 1.362)	0.144
Difference in preintervention trends (*β*_5_)	1.001	(0.981, 1.021)	0.928
Difference in postintervention step change (*β*_6_)	1.535	(1.067, 2.210)	0.021
Difference in postintervention trends (*β*_7_)	0.902	(0.796, 1.023)	0.108

^¥^ The parameters were obtained for a segmented regression model with the following structure: $Y_{t}=\beta_{0}+\beta_{1}T_{t}+\beta_{2}X_{t}+\beta_{3}X_{t}T_{t}+\beta_{4}Z+\beta_{5}ZT_{t}+\beta_{6}ZX_{t}+\beta_{6}ZX_{t}T_{t}+\epsilon_{t}$. Here, *Y_t_* is the outcome measure along time *t*; *T_t_* is the monthly time counter; *X_t_* indicates pre- and postintervention periods, *Z* denotes the intervention cohort, and *ZT_t_*, *ZX_t_*, and *ZX_t_T_t_* are interaction terms. *β*_0_ to *β*_3_ relate to the control group as follows: *β*_0_, intercept; *β*_1_, preintervention trend; *β*_2_, postintervention step change; *β*_3_, postintervention trend. *β*_4_ to *β*_7_ represent differences between the control and intervention districts: *β*_4_, difference in baseline intercepts; *β*_5_, difference in preintervention trends; *β*_6_, difference in postintervention step changes; *β*_7_, difference in postintervention trend. ^‡^ IRR is based on a log-linear GEE Poisson regression with correlation structures, as determined by the Cumby–Huizinga test and Quasi-Information Criteria; ^Þ^, Wald test.

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
