# Peer review of "Optimizing Active Tuberculosis Case Finding: Evaluating the Impact of Community Referral for Chest X-ray Screening and Xpert Testing on Case Notifications in Two Cities in Viet Nam"

_tropicalmed, 2020, doi:10.3390/tropicalmed5040181_

Round 1
Reviewer 1 Report
Dear Authors,
PFA the pdf file with my comments using the sticky notes option. I have also highlighted the relevant sections where i have provided comments.

Author Response
Dear Reviewer,
We sincerely thank you for your consideration of the research article entitled “Optimizing active tuberculosis case finding: Evaluating the impact of community referral for chest x-ray screening and Xpert testing on case notifications in two cities of Viet Nam.”
We further thank you for your thorough review and thoughtful comments, and believe that by addressing the points raised we have been able to improve the quality and readability of the manuscript. Below we have provided a point-by-point response. We have structured the feedback along the line numbers containing your feedback in the pdf file. Line numbers in our explanation correspond to those in the revised version of the manuscript showing all tracked changes.
- Line 29: We have edited the phrase to add “for active case finding” as suggested.
- Line 35-37: We did not see a comment and have skipped this highlight for the time being
- Line 39-41: We have removed the comparison as requested.
- Line 43: We have added the “for active case finding” as suggested.
- Line 52: The phrase in the parentheses was to show the formula by which treatment coverage is calculated in the WHO Global Tuberculosis Report. However, to avoid confusion we have removed the phrase in the parenthesis.
- Line 68-69: To clarify the meaning in the sentence, we have edited it to say “In Viet Nam, these barriers are exacerbated by the resource needs to compensate for lower-than-expected treatment coverage following the second prevalence survey as well as delays in the transition of TB program financing to national social health insurance.”
- Line 88: We have removed the sentence and integrated the objectives with the study design section as suggested in the next comment.
- Line 89: We have added a section called study design as suggested and inserted a statement that says: “This study was a prospective controlled cohort intervention study that aimed to measure yield, treatment outcomes and additional impact on case notifications of an intensified community-based active case finding intervention.”
- Line 136: We have inserted a flow diagram as figure 2 as suggested.
- Line 149: We would like to respectfully request that we maintain Figure 2 the way it is as the number of persons recorded as screened verbally represented those eligible and referred for CXR screening. We have made edits in the text in line 168 to reflect that the cascade presented started “with those verbally screened and eligible for CXR until treatment completion.” We did not systematically record the number of people screened and not referred for CXR screening as CXR screening represented the screening step of interest in this study. With respect to the eligibility and conversion for other indicators, the number of persons with abnormal CXR represents those eligible for Xpert, which is provided in table 1.
- Line 152-155: The unit of analysis was TB case notifications per 100,000 population, which we aggregated to intervention and control groups in pre- and post-intervention periods.
- Line 165: We have edited the text to expand on the relevant approvals obtained for the study protocol and ethical considerations.
- Line 167: Unfortunately, we did not systematically record persons declining to be included in the study, so that we are unable to provide this information.
- Line 171: We respectfully prefer not to create another column or bar chart for the cascade as much of the information would overlap with figure 2 and has been provided in table 1.
- Line 172: As previously indicated, the number of persons recorded as verbally screened comprises the number of persons eligible and referred for CXR screening.
- Line 173: As suggested, we have edited the text to include the number of persons tested by Xpert and the proportion of these persons tested among those eligible.
- Line 175: To emphasize the need for ensuring completion of the TB care cascade, we believe that ACF yield (and NNS) should be defined as those linked to treatment rather than stopping at those detected with TB. This figure is also more conservative. As such, we would respectfully request to retain the currently calculated yield and NNS.
- Line 179: We have edited all figure labels as requested.
- Line 187: We wanted to show the percentage of TB cases linked to treatment as a % of total verbally screened and referred and in order to avoid confusing readers with different denominators or to have to footnote each percentage, we prefer to present the data in the current structure.
- Line 191: As discussed above and edited in line 168, the figure of persons verbally screened indicates those eligible and referred for CXR screening.
- Line 197: We have edited the text to mention “among all sputum tests.”
- Line 224-225: We have moved the sentences “The time series data consisted of 21 observations of quarterly TB notification rates in intervention and control areas. This included 16 pre-intervention quarters and 5 quarters of implementation of ACF interventions.” to the section 2.6 lines 175-177. Table 2 includes 5 intervention quarters matched to 5 pre-intervention quarters to account for seasonality, i.e., Q1 2018 matched with Q1 2017, Q2 2018 matched with Q2 2017, Q3 2018 matched with Q3 2017, Q4 2018 matched with Q4 2017 and Q1 2019 matched with Q1 2017 again. To improve clarity, we have added this as a footnote to the table. The analysis shown in the table is based on the method developed by Blok et al. (2014) as employed by the Stop TB Partnership and accounts for crude additionality, which does not incorporate trends. The ITS analysis does include trends for which we subsequently included a 16-quarter pre-intervention period.
- Blok, L.; Creswell, J.; Stevens, R.; Brouwer, M.; Ramis, O.; Weil, O.; Klatser, P.; Sahu, S.; Bakker, M.I. A pragmatic approach to measuring monitoring and evaluating interventions for improved tuberculosis case detection. Int. Health 2014, 6, 181–188, doi:10.1093/inthealth/ihu055.
- Line 248: We do not think we should include a paragraph on cost effectiveness into the discussion since the study and presented results do not include any data on costs and therefore a discussion on this topic would be shallow and insufficient in new information. However, to accommodate the reviewer’s suggestion, we have added in lines 371-374 the statement: “Similarly, as high-burden countries begin to expand Xpert coverage through strategies such as the 2X diagnostic algorithm in Viet Nam, the cost effectiveness of these strategies need further investigation to complement their potential epidemiologic benefits.”
- Line 253: Thank you for recommending this interesting study. We have added it to the sources cited in the discussion for reference.
- Line 255: We have edited the sentence to remove the comparison.
- Line 292: Thank you and yes we concur. Moreover, community-based ACF has also been shown to reduce patient costs, further strengthening the case for scale-up of ACF.
- Line 327: We have made the editorial change as suggested.
Thank you very much once again for your review and assessment of our manuscript.
Sincerely and on behalf of the study team,
Luan Vo
Reviewer 2 Report
This is an important paper in the "End TB" era, especially given that millions of people who are estimated to develop TB each year are never diagnosed or treated for their disease. The authors present a well-done study looking at a multi-pronged approach to improving TB case finding in VietNam, which included CHW screenings for household and neighborhood contacts of persons with TB and other high-risk populations. Depending on the risk factors, persons had symptom screening, CXR, and some form of bacteriologic assessment. The paper presents interesting findings and a thorough discussion. I have some small comments that I think will strengthen the paper and which the reviewers should address.
In terms of the methods, the authors need to better define why they chose the risk groups they did.
They also need to describe why they used smear microscopy for people with normal CXR findings. Smear is not an ideal technology and is no longer recommended by the WHO as the primary diagnostic test for TB. I understand Xpert was not widely used in Viet Nam at this time. However, persons with CXR abnormalities were likely to have a higher bacillary burden and thus may have been more likely to be picked up on smear microscopy. It would help to understand the authors' rationale for using a less sensitive test in people without CXR findings.
The authors also need to describe how children were screened and what might have been done differently for them.
While I understand the inclusion of people with prior TB given their risk factor of developing a second episode, the use of Xpert MTB/RIF is not recommended in this population and the authors should justify why they used Xpert in persons with previous TB.
The authors should also comment on whether or not HIV testing was offered/done and if not, why not since HIV is a significant risk factor for TB. I realize VietNam is a low-prevalence HIV country, but this information would still be important to include.
In the discussion section of the paper, it would be good if the authors could include a sentence or two about the challenges they found (if any) with CXR interpretation. Did they consider using automated systems to interpret the CXRs?
The authors should also discuss the limits of focusing on the NNS (number needed to screen) measure of effectiveness. I realize they include it here because it is commonly presented in studies such as this. However, NNS can be problematic. For example, if a high risk group is screened the NNS may be low, but if the number of people in the high risk group is low, not many of the "missing" people will be found. The authors' own data show this, as most of the people living with TB who were diagnosed in this study in terms of absolute numbers were not from the highest risk group. To find "the missing" with TB will require the engagement of people from lower risk groups who represent a broader segment of the population, and the authors should discuss this as well as the limitations of NNS in understanding a study like this and how to "find the missing" millions.
Finally, while I realize it was beyond the scope of this paper, the authors should consider mentioning whether or not preventive therapy was offered to any household contacts or other high-risk groups who did not have active TB.
Author Response
Dear Reviewer,
We sincerely thank you for your consideration of the research article entitled “Optimizing active tuberculosis case finding: Evaluating the impact of community referral for chest x-ray screening and Xpert testing on case notifications in two cities of Viet Nam.”
We further thank you for your thorough review and thoughtful comments, and believe that by addressing the points raised we have been able to improve the quality and readability of the manuscript. Below we have provided a point-by-point response. Wherever appropriate, we have provided line numbers of the edits. These numbers correspond to those in the revised version showing all tracked changes.
- In terms of the methods, the authors need to better define why they chose the risk groups they did.
- We have added the following sentence to lines 130-132 to further clarify the selection of the target groups: “These groups were prioritized at the discretion of the Provincial TB control programs based on the historically observed higher incidence in these groups.”
- They also need to describe why they used smear microscopy for people with normal CXR findings. Smear is not an ideal technology and is no longer recommended by the WHO as the primary diagnostic test for TB. I understand Xpert was not widely used in Viet Nam at this time. However, persons with CXR abnormalities were likely to have a higher bacillary burden and thus may have been more likely to be picked up on smear microscopy. It would help to understand the authors' rationale for using a less sensitive test in people without CXR findings.
- We completely concur with the reviewer’s comment and want to emphasize the point that Viet Nam does not yet have the capacity to extend Xpert testing to every person with suspected TB, as noted by the reviewer.
- As such, the current national diagnostic roadmap includes X-ray as a triage for two main reasons: 1) to improve the cost effectiveness of the introduction of Xpert as the initial diagnostic test; 2) due to the high rate of asymptomatic TB found on the country’s two prior prevalence surveys.
- The innovation here is not to focus on symptomatic persons without a CXR or with normal CXR results (which historically has shown low yields), but to keep in mind that the 2X diagnostic algorithm introduced in this study is agnostic of symptomatic presentation.
- Other than that, the NTP’s diagnostic roadmap does indeed include the plan to expand Xpert testing to all persons with suspected TB in the future once sufficient resources have been mobilized.
- The authors also need to describe how children were screened and what might have been done differently for them.
- We have added a sentence in line 160-161 on children with suspected TB: “Children with presumptive TB were referred to the Provincial Lung Hospital for further evaluation.”
- While I understand the inclusion of people with prior TB given their risk factor of developing a second episode, the use of Xpert MTB/RIF is not recommended in this population and the authors should justify why they used Xpert in persons with previous TB.
- This is a very astute observation and justified question given that we omitted this process as it is part of the standard national diagnostic algorithm.
- For patients with a history of TB and a positive Xpert result, an immediate smear is performed. If that is positive and the Xpert test was rifampicin-sensitive, the patient is enrolled onto DS-TB treatment. If the smear is negative, a culture is performed to ascertain presence of live bacteria.
- We have reflected that by adding a sentence to lines 157-158 stating: “Persons with a positive Xpert result and a history of TB were further tested on smear and culture, if needed, as per NTP guidelines.”
- The authors should also comment on whether or not HIV testing was offered/done and if not, why not since HIV is a significant risk factor for TB. I realize Viet Nam is a low-prevalence HIV country, but this information would still be important to include.
- While we concur with the reviewer that HIV is a significant risk factor, there are two considerations in favor of omitting this aspect.
- First, HIV testing is provided as part of routine NTP operations as per national TB guidelines.
- Second, Viet Nam is a low HIV burden country with a concentrated epidemic and HIV was not a focus of the project, so that we respectfully propose that we do not reiterate routine programmatic activities.
- In the discussion section of the paper, it would be good if the authors could include a sentence or two about the challenges they found (if any) with CXR interpretation. Did they consider using automated systems to interpret the CXRs?
- Thank you for this comment as we certainly concur with the potential for heterogeneity in CXR interpretation and a possible application of Computer-Aided Reading (CAR) solutions.
- With that said, we are in the process of finalizing an analysis of 14 different CAR solutions in comparison with a human expert reader (15+ years of experience) and human field reader (5+ years of experience), which will be published in the near future.
- This analysis showed that none of the CAR existing solutions significantly outperformed the human expert and only one very recent version outperformed a human field reader.
- Nevertheless, we have made an addition to the discussion in lines 355-358 stating: “Another limitation of the approach consisted of the potential heterogeneity in CXR interpretation across different human readers. Future studies should further explore this limitation and the utility of computer-aided reading solutions in active case finding settings such as that implemented on this project [53].”
- 53. Khan, F.A.; Pande, T.; Tessema, B.; Song, R.; Benedetti, A.; Pai, M.; Lönnroth, K.; Denkinger, C.M. Computer-aided reading of tuberculosis chest radiography: Moving the research agenda forward to inform policy. Eur. Respir. J. 2017, 50, doi:10.1183/13993003.00953-2017.
- The authors should also discuss the limits of focusing on the NNS (number needed to screen) measure of effectiveness. I realize they include it here because it is commonly presented in studies such as this. However, NNS can be problematic. For example, if a high risk group is screened the NNS may be low, but if the number of people in the high risk group is low, not many of the "missing" people will be found. The authors' own data show this, as most of the people living with TB who were diagnosed in this study in terms of absolute numbers were not from the highest risk group. To find "the missing" with TB will require the engagement of people from lower risk groups who represent a broader segment of the population, and the authors should discuss this as well as the limitations of NNS in understanding a study like this and how to "find the missing" millions.
- We again thank the reviewer for this highly insightful and valid comment. We fully concur and in fact have a manuscript in print elaborating this point based on ACF work in HCMC.
- For this manuscript, we have made the following edits and additions to lines 306-310 to address the reviewer’s point: “However, studies have repeatedly shown that high yield and low NNS from more sensitive diagnostic tools and algorithms in isolation does not necessarily translate to additional notifications [31,32]. As vulnerable groups with low NNS often are also small in numbers, but that it has been noted that the use of more sensitive diagnostic tools and algorithms must be complemented by efforts to increase the number of people referred for screening and testing [33].”
- 31. Theron, G.; Peter, J.; Dowdy, D.; Langley, I.; Squire, S.B.; Dheda, K. Do high rates of empirical treatment undermine the potential effect of new diagnostic tests for tuberculosis in high-burden settings? Lancet Infect. Dis. 2014, 14, 527–532, doi:10.1016/S1473-3099(13)70360-8.
- 32. Theron, G.; Zijenah, L.; Chanda, D.; Clowes, P.; Rachow, A.; Lesosky, M.; Bara, W.; Mungofa, S.; Pai, M.; Hoelscher, M.; et al. Feasibility, accuracy, and clinical effect of point-of-care Xpert MTB/RIF testing for tuberculosis in primary-care settings in Africa: A multicentre, randomised, controlled trial. Lancet 2014, 383, 424–435, doi:10.1016/S0140-6736(13)62073-5.
- 33. Creswell, J.; Rai, B.; Wali, R.; Sudrungrot, S.; Adhikari, L.M.; Pant, R.; Pyakurel, S.; Uranw, D.; Codlin, A.J. Introducing new tuberculosis diagnostics: The impact of Xpert MTB/RIF testing on case notifications in Nepal. Int. J. Tuberc. Lung Dis. 2015, 19, 545–551, doi:10.5588/ijtld.14.0775.
- Finally, while I realize it was beyond the scope of this paper, the authors should consider mentioning whether or not preventive therapy was offered to any household contacts or other high-risk groups who did not have active TB.
- This is also a much appreciated and relevant comment. We have added the following sentence to lines 370-371 to reflect our concurrence: “Lastly, the study focused mainly on active tuberculosis and future studies should investigate the possibility of integrated screening, testing and treatment of TB and latent TB infection”.
Thank you very much once again for your review and assessment of our manuscript.
Sincerely and on behalf of the study team,
Luan Vo
Reviewer 3 Report
The study evaluated the results of the screening activities using chest X-ray and Xpert in collaboration with community health workers. I think this study deserves to be published, however, I would like to raise the following concerns.
- Title: treatment notifications -> tuberculosis or case notifications
- Study setting: How were the controls selected? The rationale seems insufficient in the manuscript.
- ACF activities & Figure 3: What is the rationale for using Xpert among TB suggestive CXR abnormalities and smear microscopy among non-TB suggestive CXR abnormalities. This strategy resulted in different proportions of Xpert testing across the target groups implying different yields due to different sputum tests.
- Figure 2: Care cascade is designed for evaluating the proportion of diagnosed and successfully treated TB patients among estimated TB patients. Therefore, this method should be applied as follows. No. of estimated TB patients eligible for screening -> CXR-screened TB patients -> diagnosed (sputum-tested) TB patients -> treatment-initiated TB patients -> successfully-treated TB patients where the first two indicators can be calculated retrospectively using the proportion of screening activities.
- Table 1: Why don't you divide it into process indicators (Table 1) and treatment outcomes (Table 2)
Author Response
Dear Reviewer,
We sincerely thank you for your consideration of the research article entitled “Optimizing active tuberculosis case finding: Evaluating the impact of community referral for chest x-ray screening and Xpert testing on case notifications in two cities of Viet Nam.”
We further thank you for your thorough review and thoughtful comments, and believe that by addressing the points raised we have been able to improve the quality and readability of the manuscript. Below we have provided a point-by-point response. Wherever appropriate, we have provided line numbers of the edits. These numbers correspond to those in the revised version showing all tracked changes.
- Title: treatment notifications -> tuberculosis or case notifications
- We have edited the title to state “case notifications”
- Study setting: How were the controls selected? The rationale seems insufficient in the manuscript.
- We have added a sentence to lines 105-108 to expand on the rationale for selection of the control area stating: “The control area was selected based on recommendations from the provincial TB programs to match the intervention areas in approximate size and TB burden as well as absence of any case finding interventions.”
- ACF activities & Figure 3: What is the rationale for using Xpert among TB suggestive CXR abnormalities and smear microscopy among non-TB suggestive CXR abnormalities. This strategy resulted in different proportions of Xpert testing across the target groups implying different yields due to different sputum tests.
- We completely concur with the reviewer’s comment and want to emphasize the point that Viet Nam does not yet have the capacity to extend Xpert testing to every person with suspected TB.
- As such, the current national diagnostic roadmap includes X-ray as a triage for two main reasons: 1) to improve the cost effectiveness of the introduction of Xpert as the initial diagnostic test; 2) due to the high rate of asymptomatic TB found on the country’s two prior prevalence surveys.
- The innovation here is not to focus on symptomatic persons without a CXR or with normal CXR results (which historically has shown low yields), but to keep in mind that the 2X diagnostic algorithm introduced in this study is agnostic of symptomatic presentation.
- Other than that, the NTP’s diagnostic roadmap does indeed include the plan to expand Xpert testing to all persons with suspected TB in the future once sufficient resources have been mobilized.
- Figure 2: Care cascade is designed for evaluating the proportion of diagnosed and successfully treated TB patients among estimated TB patients. Therefore, this method should be applied as follows. No. of estimated TB patients eligible for screening -> CXR-screened TB patients -> diagnosed (sputum-tested) TB patients -> treatment-initiated TB patients -> successfully-treated TB patients where the first two indicators can be calculated retrospectively using the proportion of screening activities.
- We would like to respectfully request that we maintain Figure 2 the way it is as all of the requested indicators have been included in the chart.
- To clarify the number eligible for CXR, we have edited the text in line 168 to reflect that the cascade presented started “with those verbally screened and eligible for CXR until treatment completion.” With respect to the eligibility and conversion for other indicators, the number of persons with abnormal CXR represents those eligible for Xpert, which is provided in table 1.
- We did not systematically record the number of people screened and not referred for CXR screening as CXR screening represented the screening step of interest in this study.
- Regarding the other requested indicators in the cascade, they are reflected in columns 2, 4/5, 6 and 7 of the figure.
- Table 1: Why don't you divide it into process indicators (Table 1) and treatment outcomes (Table 2)
- We have split the table into two as requested by the reviewer and have added a short description of the treatment success rates in lines 213-216.
Thank you very much once again for your review and assessment of our manuscript.
Sincerely and on behalf of the study team,
Luan Vo
Round 2
Reviewer 1 Report
The authors have addressed the comments to my satisfaction.